# MULTI TASK LEARNING OF DIFFERENT CLASS LABEL REPRESENTATIONS FOR STRONGER MODELS

## ABSTRACT

We find that the way in which class labels are represented can have a powerful effect on how well models trained on them learn. In classification, the standard way of representing class labels is as one-hot vectors. We present a new way of representing class labels called Binary Labels, where each class label is a large binary vector. We further introduce a new paradigm, multi task learning on different label representations. We train a network on two tasks. The main task is to classify images based on their one-hot label, and the auxiliary task is to classify images based on their Binary Label. We show that networks trained on both tasks have many advantages, including higher accuracy across a wide variety of datasets and architectures, both when trained from scratch and when using transfer learning. Networks trained on both tasks are also much more effective when training data is limited, and seem to do especially well on more challenging problems.

## 1 INTRODUCTION

All supervised learning problems involve three components, data, a model, and labels. A tremendous amount of work has been done involving the first two parts, models, and data, but labels have been mostly ignored. Deep learning architectures get ever more complicated (Villalobos et al., 2022), and many excellent techniques exist to learn strong representations of the data that is fed into the model (He et al., 2022a; Chen et al., 2020), but the labels themselves remain as simple as they were 20 years ago. Let's illustrate this point with a common deep learning task, image classification on ImageNet (Deng et al., 2009). The leaderboard is full of algorithms that employ powerful unsupervised pre-training methods to learn representations of the data, and make use of massive model architectures with hundreds of layers and millions of weights. But if one surveys the top state of the art models, the target labels are all simple one-hot encoded vectors (Yu et al., 2022; Wortsman et al., 2022; Chen et al., 2022).

Recently, various alternative label representations have been proposed, where labels are represented as dense vectors (Chen et al., 2021). While on some metrics, such as robustness and data efficiency, these labels were able to outperform standard label representations, on other key metrics, such as accuracy, they performed worse than one-hot labels. Additionally, these dense labels were slower at inference time, for two reasons. First, because they added more weights to the networks that used them, the forward pass takes slightly longer. Second, and more significant, is that classification on dense labels is slower by definition since it involves comparing the network output with each label to find the class with the nearest label, as opposed to one-hot labels with a softmax output where the classification is done directly. Furthermore, these dense labels were much slower to converge, often taking four times as many epochs. As such, choosing between these label representations is essentially a trade off between opposing factors.

In this paper we explore the key idea of: given various ways of representing labels, why only choose one? Every way of representing labels can become a different task for the network to learn. By choosing to represent the same class labels in different ways, we are reusing the supervision to create multiple learning tasks. We thus propose using recent work in the field of multi task learning to train networks to recognize multiple representations of the same labels. We find that doing so allows us to mitigate the tradeoff above.

From an intuitive perspective, it makes sense that learning multiple representations of the same concept can be useful. To give an analogy from data structures, while linked lists and arrays represent

the same underlying data, each data structure has inherent advantages and disadvantages, and some algorithms will rely on redundant data storage of both types to get the advantages of both. While it may be novel to study explicit redundant label representations, they commonly exist implicitly. Research on the human brain indicates we make heavy use of redundant representations (Pieszek et al., 2013), and artificial neural networks are already doing this implicitly (Doimo et al., 2021).

We present a new label representation type, Binary labels, where each class label is represented as a binary vector. This representation has key properties that led us to think it would create a useful auxiliary task to improve accuracy, and indeed our experiments verify this.

We make the following main contributions:

1. We present the novel paradigm of using several label representations of a single label to augment our network supervision and create auxiliary tasks for the network to learn
2. We present a new label representation type, Binary Labels
3. We present results that demonstrate the strength of our approach

We hope this will inspire further research on this topic.

## 2 RELATED WORK

We unite work from two areas, label representation, and multi task learning.

### 2.1 LABEL REPRESENTATION

There are many scenarios where it makes sense for a network performing image classification to output a dense representation as opposed to a softmax vector. For example, in few-shot learning, a common approach, often termed embedding learning, is to learn a lower dimensional representation of the input, such that similar images are near each other in the embedding space (Wang et al., 2020a). It is also common to use autoencoders to try and learn image representations, which usually involves some sort of dense intermediate representation that is used to reconstruct the original (Pinaya et al., 2020). While both of these methods involve networks that output a representation, they do not make any attempt to generate or make use of alternative label representations. Indeed, relatively little research has been conducted on the labels themselves or more specifically, their representation, aside from (Chen et al., 2021), where several alternative methods were proposed. They reported gains on robustness and data efficiency. However convergence was much slower, and accuracy was slightly lower than when using standard softmax labels. Aside form that, the closes work is label smoothing (Szegedy et al., 2016; Sun et al., 2017), although as (Chen et al., 2021) note, it is quite different.

### 2.2 MULTI-TASK LEARNING

Multi task learning, first proposed in 1997 (Caruana, 1997), has gained traction as a powerful way of training a single network to learn multiple tasks at the same time (Crawshaw, 2020; Ruder, 2017; Zhang & Yang, 2021; Vandenhende et al., 2021). Often times, doing so leads to gains on both tasks, and recent work has explored what sorts of tasks are best learned together (Standley et al., 2020). Auxiliary learning is a subtopic in multi task learning that deals specifically with learning a main task, which we care about, and auxiliary tasks, whose only purpose is to increase performance for the main task (Vafaeikia et al., 2020; Liebel & Körner, 2018).

We make use of several standard tools from multitask learning. We use a shared trunk approach (Crawshaw, 2020), such that all tasks share a backbone network, but have different output heads. When combining the losses of different tasks, we explored several popular methods including PC-Grad (Yu et al., 2020), GradNorm (Chen et al., 2018), GradVac (Wang et al., 2020b), and Mtadam (Malkiel & Wolf, 2020), but ultimately settled on Metabalance (He et al., 2022b), which is specifically aimed at auxiliary learning [1]. We also assign weights to each task, such that the sum of all weights on all tasks sum to 1.

---

[1] We believe we caught a small mistake in the Metabalance paper, and notified the author. The version we use is slightly different in that it requires two forward passes to correctly compute the gradients

While we make use of and build off of work in the field of Multi Task Learning, we are unaware of any work that tries to learn different representations of the same task simultaneously. The closest work, supervised autoencoders (Le et al., 2018), attempts to learn classification and image reconstruction at the same time. While this is an example of automatically generating an auxiliary task, and applying the tools of multi task learning, the auxiliary task is not simply a different representation of the main classification task, but something entirely different (input reconstruction).

Another related method is Semantic Softmax (Ridnik et al., 2021), which uses a hierarchical dataset to generate one-hot softmax labels for each level of the hierarchy, and trains on them together. However, this method requires additional supervision: the class hierarchy needs to be given, and in most datasets, we are given no such hierarchy. Furthermore, the various labels are learning to classify at different levels of the hierarchy, and thus are distinct classification tasks all represented as one-hot softmax vectors, as opposed to our work where each classification task is identical, just using different label representations.

## 3 MULTI TASK LEARNING OF MULTIPLE REDUNDANT LABEL REPRESENTATIONS

It is well known that learning multiple tasks, when the two tasks are deeply related, often leads to improved performance on both tasks (Standley et al., 2020). It thus makes sense that training a network to learn two different label representations of the same class would be useful. In this case the two tasks are not just related, they are identical, with the only difference being the chosen label representation method.

Lets define our approach more formally. First, we will start with the standard way of doing things. Lets say we are given a labeled dataset where there are $n$ classes, where each class label is provided as a number in the range of $0$ to $n-1$. Say we have a dataset of $I$ samples, where $X_i, y_i$ denotes the $i$th input and label respectively.

We define some function $f$ that maps these labels to the representations we would like the network to learn. In the standard approach, this simply is a one-hot encoding, but in principle this function can be anything. We further define a loss function $L$, and a network $N$.

The goal is to minimize the total loss, given as:

$$\sum_{i=0}^{I} L[N(X_i), f(y_i)]$$

This is how networks are normally trained. The only thing noteworthy is that we explicitly defined $f$, the function that converts the class label to the one-hot representation that the network learns.

In the redundant representation multi task learning setting that we propose, we extend the above. We may have several such functions $f_0 - f_{T-1}$, where $T$ is the number of tasks we are trying to learn. Each function takes in a number, in range $0$ to $n-1$ representing the class, and produces a label.

Correspondingly, our network has one backbone $N$, with $T$ output heads $h_0 - h_{T-1}$. We will use $N_{h_t}(X_i)$ to denote the output of the $t$th head on input $X_i$.

We also define $T$ loss functions, $L_0 - L_{T-1}$, with each one corresponding to a task.

The goal is to minimize the loss on all of these tasks, given as:

$$\sum_{i=0}^{I} \left( \sum_{t=0}^{T-1} L_t(N_{h_t}(X_i), f_t(y_i)) \right)$$

Thus, instead of training the network to recognize just one representations, we may utilize several representations together, learning them all at the same time by treating them as different tasks. The tasks all share the main network parameters, but each task has a different head that learns the

representation specific to it. In this way we are reusing our single supervision source to generate many related tasks.

We further can define $T$ task weights, $w_0$ to $w_{T-1}$, with the constraints that all weights are positive, and that $\sum_{t=0}^{T} w_t = 1$. The loss we want to minimize then becomes:

$$\sum_{i=0}^{I} \left( \sum_{t=0}^{T-1} w_t * L_t(N_{h_t}(X_i), f_t(y_i)) \right)$$

It should be noted, that although the high level goal is to minimize the above, when calculating the gradients for backpropogation, often times better results are achieved by using more clever combination techniques than addition (Crawshaw, 2020).

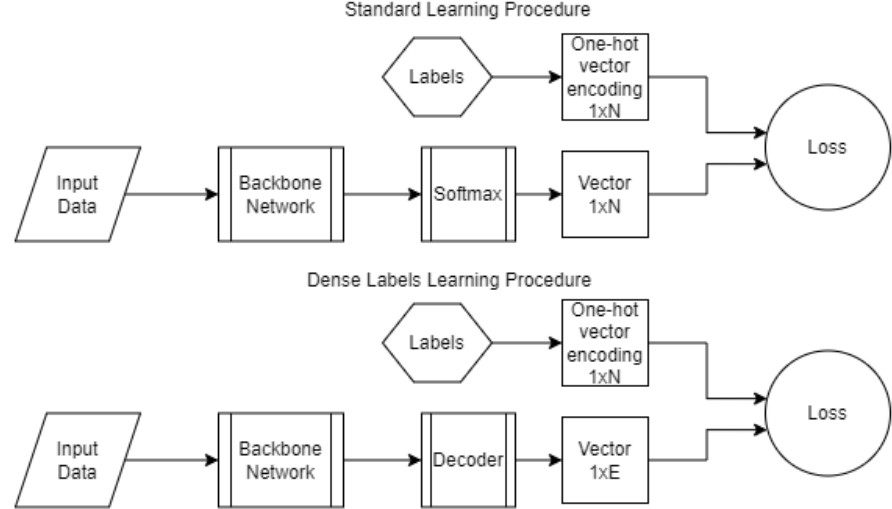

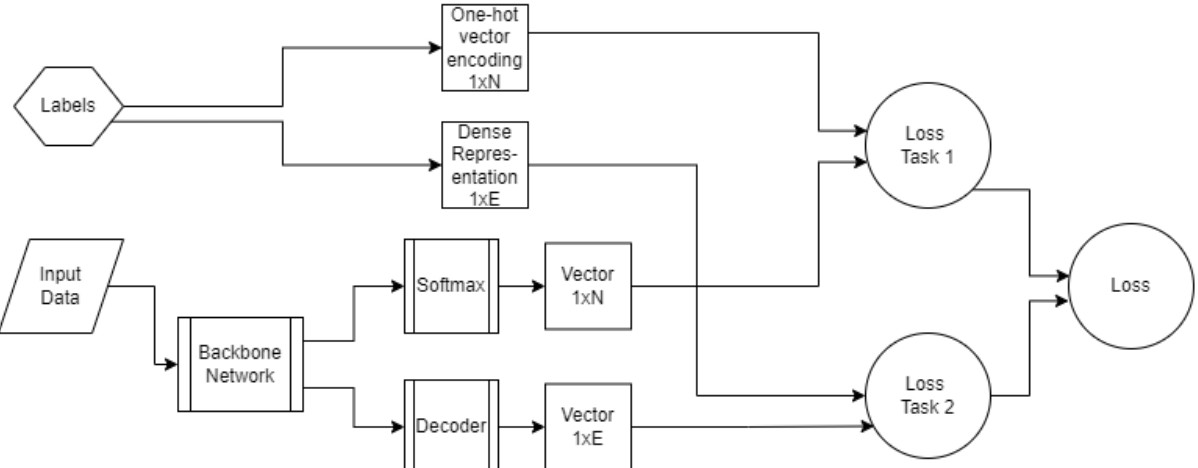

Figure 1: Multi task learning on two label representations

To make this an auxiliary learning problem, we can define one task, $t_0$ as the main task, and the rest as auxiliary. After training is finished, we can discard all the heads of the auxiliary tasks, and are left with a network with one output head trained on $t_0$.

Figure 1 compares a softmax classification architecture, a dense label classification architecture, and a multi task learning architecture training on 2 representations, one softmax, one dense.

## 4 BINARY LABELS

Binary labels, as the name implies, are vectors of bits, with each value being either a $0$ or a $1$. Thus, each class is represented as a binary vector of length $E$, where $E$ is the embedding dimension.

While Binary Labels might seem like a strange choice to represent class labels, there are actually intuitive factors that motivate this representation as a useful auxiliary task to improve accuracy.

To illustrate why this representation is useful, consider a single dimension of the embedding space. For this dimension, each class label will either be a $0$, or a $1$, by definition. This splits the classes into two groups, the group of classes that were labeled $1$ in this dimension, and the group labeled $0$. What this means is that the neuron that learns to output this dimension of the label space is being trained on a binary classification task, where the classes in the dataset have been split into two groups. Now lets consider the binary labels as a whole. For a set of labels of dimension $E$, the network is being trained on $E$ binary classification tasks, each one potentially of a different split of the classes. Another way of thinking about this is that each bit is learning the commonalities between a different set of classes that are grouped together

Intuitively, having each dimension solve a different binary classification problem makes sense as an auxiliary task, since we can expect the network to learn useful features that distinguish between subgroups of classes that may not have been learned otherwise.

The number of ways we can divide all classes into 2 groups increases exponentially with the number of classes. For example, if we only have four classes, A B C D, the following binary labels would cover all 7 possible divisions (see figure 2).

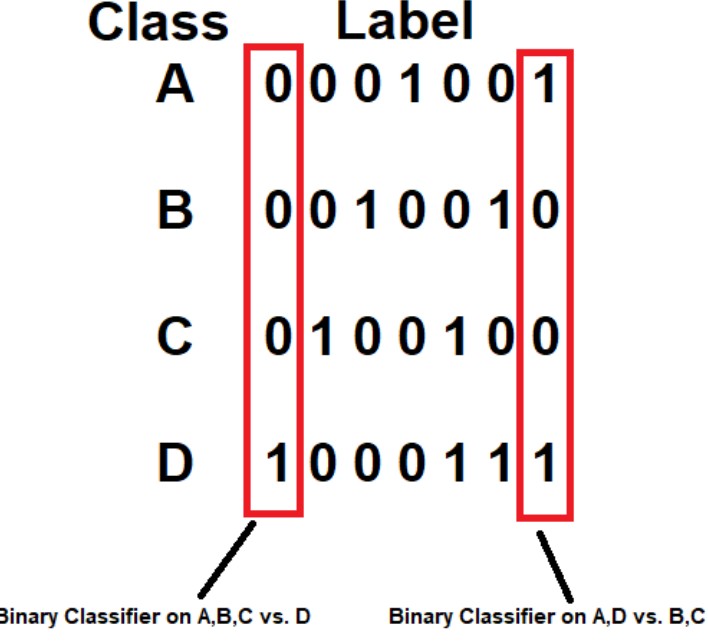

Figure 2: Binary Labels

Note, each column represents the same division even if all the bits are inverted.

In general, for $N$ classes, we can divide them into $\binom{N}{\frac{N}{2}}$ evenly split groups, and if we also consider unevenly split groups, this becomes $\sum_{k=1}^{N/2} \binom{N}{K}$ groups.

If $N$ is small, it is feasible to use all of these divisions. As seen in figure 2, when $N$ is 4, there are only 7 such groups, 3 of which are evenly split. When $N$ is 10, there are 252 evenly split groups, and 637 total groups.

In these cases, when $N$ is small, the binary labels could be carefully constructed to utilize every division. However, as N gets large, this is infeasible. For example, with 20 classes, we already have 184756 evenly split divisions, far too many for us to attempt to learn each one.

Thus, for a large number of classes, we wish to only sample from the number of possible divisions. One very easy way of doing this is by simply randomly generating the binary labels, where each bit in each label has equal probability of being a $0$ or $1$.

In summary, the motivation behind using Binary labels as an auxiliary task is that it teaches the network to distinguish between the classes in many different ways, and this may provide useful features that can increase accuracy on the main task.

## 5 EXPERIMENTAL SETUP

We ran three sets of experiments. In the first, we demonstrate the usefulness of this method when training from scratch. We then demonstrate that our method leads to accuracy gains when performing transfer learning as well. Finally, we analyze data efficiency, and compare Binary Labels with other auxiliary label representations.

### 5.1 DATASETS

For the training from scratch experiments, we evaluate our model on five datasets, Cifar-100 (Krizhevsky et al., 2009), Caltech101 (Fei-Fei et al., 2004), Sun397 (Xiao et al., 2010), FGVC-Aircraft (Maji et al., 2013), and Stanford Cars (Krause et al., 2013). Stanford Cars is particularly interesting since Resnets do surprisingly poorly on this dataset when trained from scratch [2].

### 5.2 MODELS

On Cifar-100, we evaluated three models, VGG19 (Simonyan & Zisserman, 2014), ResNet32, and ResNet110 (He et al., 2016). This choice was mostly made because we wanted to reproduce and improve upon the experiments of (Chen et al., 2021), where exactly these networks were used on Cifar-100. For the other datasets, we used Resnext-50 (Xie et al., 2017), Resnet-50 and Resnet-101 (He et al., 2016).

For the transfer learning experiments, we used Caltech101 and FGVC-Aircraft, and evaluated on Resnet-50, Resnet-101, VIT-B16 (Dosovitskiy et al., 2020), Beit-B16 (Bao et al., 2021), and MLP-Mixer-B16 (Tolstikhin et al., 2021).

### 5.3 DATA EFFICIENCY

For the final experiments, where we analyzed how Binary Labels compare with other label representations in the realm of data efficiency, we used VGG19 and Resnet-110, trained on Cifar-100, once again to be consistent with (Chen et al., 2021).

### 5.4 MULTI TASK ARCHITECTURE AND SETUP

In all of our multi-task experiments, we trained the network on two tasks. The main task was to predict the standard softmax one-hot labels, and the auxiliary task was to predict Binary Labels. While the exact value of $E$, the embedding dimension for the Binary Labels, is somewhat arbitrary,

---

[2] While we did not find anything in the literature pointing this out, several open source implementations of Resnets on Stanford Cars confirm this. See for example https://github.com/eqy/PyTorch-Stanford-Cars-Baselines

in all of our experiments in this paper, $E$ was equal to 4096, so that we would be directly comparable to (Chen et al., 2021), where labels were given as 64x64 matrices.

Each network that we trained had to be modified for the multi-task setting. We kept the standard softmax output as the head that predicts one-hot labels, which we defined as the main task. For predicting Binary Labels, the auxiliary task, we appended to each network a dense label head. After training was done, this head was discarded, and the final networks that were produced are architecturally equivalent to the standard architectures, and so all comparisons are completely fair. (This also means that at inference time, our networks are just as fast as the baseline). Accuracy is reported on the main task. The dense label head is similar to the decoders used in (Chen et al., 2021), except that we added a single fully connected layer to the end. For loss, we used cross entropy for the softmax heads, and binary cross entropy for the binary label heads.

There is also an important hyperparameter at play here, the weight we assign each task. For the experiments below, we tried two weighting schemes, and report the higher value. The two task weights used were [0.5,0.5], representing equal weighing on both tasks, and [0.9, 0.1], where we give clear priority to the main task and less priority to the auxiliary task.

## 6 RESULTS

### 6.1 TRAINING FROM SCRATCH

We present the results of the training from scratch experiments below.

The goal of these experiments was to demonstrate the value of multi task learning on redundant label representations, and specifically to demonstrate that Binary Labels generate a useful auxiliary task. For all experiments, 3 runs were averaged. We report an increase in accuracy for all architectures and datasets tested.

On Cifar-100, to be consistent with (Chen et al., 2021), we used the same models with the same hyper parameters such as learning rate, weight decay, batch size and number of epochs.

The results on Cifar-100 are below:

Table 1: Results on Cifar-100 (accuracy %)

| Network | Standard | MTL Binary Labels | error reduction |
|---|---|---|---|
| VGG19 | 70.765 | **71.57** | 2.7% |
| Resnet32 | 69.375 | **69.7975** | 1.3% |
| Resnet110 | 71.72 | **72.36** | 2.2% |

While the accuracy gains on Cifar-100 were modest, we also ran experiments on Stanford Cars, Caltech-101, Sun-397, and FGVC-Aircraft, with much larger gains. For these experiments, we used Resnet50, Resnext50, and Resnet101. We trained for 90 epochs, with an initial learning rate of 0.1 that was relaxed by 0.1 every 30 epochs. We present the results below:

Table 2: Results on four other datasets (accuracy %)

| | Dataset | | | | | | | |
|---|---|---|---|---|---|---|---|---|
| | Cars | | Caltech | | Sun | | Aircraft | |
| Network | Baseline | Ours | Baseline | Ours | Baseline | Ours | Baseline | Ours |
| Resnet50 | 13.27 | **27.11** | 63.95 | **72.63** | 27.74 | **39.26** | 56.38 | **63.52** |
| Resnext50 | 6.38 | **41.15** | 67.37 | **71.03** | 30.5 | **39.86** | 44.4 | **54.55** |
| Resnet101 | 17.29 | **31.9** | 68.55 | **72.98** | 30.58 | **37.42** | 52.66 | **68.26** |

The accuracy gains here are quite strong across a wide range of datasets and models, demonstrating that our method can be used as a general accuracy enhancement. Of particular note are the results on Stanford Cars, where the baseline networks struggled to learn anything.

We also want to note, Binary Labels were the only auxiliary label representation type we tested on that gave accuracy improvements. The other label representations from (Chen et al., 2021) lead to decreases in accuracy. We further note that Binary labels only led to accuracy improvements when trained as an auxiliary task. As a main task, or as the only task, the accuracy when using Binary Labels decreased as compared with the baseline. Thus, the benefit of Binary Labels is only realized when used in the context of our approach.

## 6.2 TRANSFER LEARNING

We also demonstrate the usefulness of our technique when using transfer learning.

All networks were pretrained on ImageNet. We used Resnet50, Resnet101, VIT-B16, Beit-B16, and MLP-Mixer, representing a large range of SOTA network architectures. We trained on Caltech-101 and FGVC-Aircraft. Training was done for 60 epochs, using the same hyperparameters as (Ridnik et al., 2021). Again, we present the average of three runs. The results are given below:

Table 3: Transfer learning results (accuracy %)

|  | Dataset | | | |
| --- | --- | --- | --- | --- |
|  | Caltech | | Aircraft | |
| Network | Baseline | Ours | Baseline | Ours |
| Resnet50 | 95.6 | **96.17** | 71.99 | **75.14** |
| Resnet101 | 96.5 | **96.9** | 71.52 | **76.45** |
| VIT-B16 | **96.62** | 96.5 | 82.97 | **83.86** |
| Beit-B16 | 96.94 | **97.07** | 84.79 | **85.9** |
| MLP-Mixer | 82.39 | **83.91** | **27.67** | 24.82 |

As these results demonstrate, our method is a useful accuracy enhancer when doing transfer learning as well. We also note that the lower the baseline accuracy is (i.e. the more challenging a task is for a given network), the more our method seems worthwhile, with the exception of MLP-Mixer on the Aircraft dataset, where both models did very poorly, but ours did worse. Once again, we want to point out that Binary Labels were the only label representation that led to accuracy improvement, and even then only when used as an auxiliary task.

## 6.3 DATA EFFICIENCY

While Binary Labels were superior to the other label representations given in (Chen et al., 2021) when it came to accuracy, we now explore another key metric: data efficiency. Data efficiency has been widely researched in the context of meta learning and few shot learning (Thrun & Pratt, 2012; Vilalta & Drissi, 2002; Vanschoren, 2018; Wang & Yao, 2019), and indeed, (Chen et al., 2021) noted that they did not see accuracy improvements when training on their label representations, but reported strong gains in data efficiency. Thus, it is only natural to see how Binary Labels fair when it comes to data efficiency. We compare them with random labels generated from a uniform distribution, since this representation achieved strong efficiency gains in (Chen et al., 2021). Results are on Cifar-100, using the same hyper parameters as above, with one difference: taks weights are [0.1,0.9], where we give more weight to the auxiliary task of learning the dense labels. We found when it comes to data efficiency, these parameters lead to stronger results.

We present results from training on 1%, 2%, 4%, and 8% of the dataset:

Table 4: Data efficiency: training on small percentages of the dataset (accuracy %)

| Percentage | 1% | | | 2% | | |
|---|---|---|---|---|---|---|
| Network | Baseline | Binary | Random | Baseline | Binary | Random |
| VGG19 | 1.86 | **9.24** | 7.93 | 4.4 | **13.03** | 12.39 |
| Resnet110 | 4.3 | **6.32** | 6.11 | 8.5 | **9.25** | 8.83 |
| Percentage | 4% | | | 8% | | |
| Network | Baseline | Binary | Random | Baseline | Binary | Random |
| VGG19 | 9.4 | **18.36** | 18.03 | 17.4 | **30** | 22.23 |
| Resnet110 | 15.31 | **16.11** | 13.05 | 25.1 | **25.61** | 23.66 |

The results are quite interesting. While both multi task methods generally did better than the baseline, Binary Labels outperformed random labels, and the difference was far more pronounced on VGG-19 than Resnet-110. Given that state of the art models rely heavily on large amounts of training data (Krizhevsky et al., 2009; Russakovsky et al., 2015; Kuznetsova et al., 2020), something generally not available for real world problems, this is a very useful property.

## 7 CONCLUSION AND FUTURE DIRECTIONS

We present a completely new paradigm: Using alternative label representations to generate auxiliary tasks that are then learned in parallel, using techniques from multi task learning. We propose Binary Labels as a simple method of obtaining an auxiliary task, and achieve strong results. Our experiments demonstrate that this is an exciting new direction.

However, this is an entirely new learning paradigm, and there remains a lot to be explored. We propose the following questions, and we think exploring them can lead to even stronger results than the ones presented in this paper:

1. What types of label representations make for good auxiliary tasks? Can we improve upon the Binary Labels presented here?

2. It is possible different representations are useful depending on the goal. Can we find representations that help with accuracy, robustness, efficiency, convergence, etc. ?

3. In this paper, we only explored using two different label representations and training on two tasks. However, it is possible a larger number would lead to better results. We would like to explore this direction as well, especially considering that different representations may help us in different ways (i.e. accuracy vs robustness).

4. We found that the benefits of the auxiliary task was very sensitive to how we combined the loss functions. What is the ideal way to do this?

5. For the dense label heads, we used the decoder from (Chen et al., 2021), with an additional fully connected layer. We tried other schemes that were not as useful. Even though these output heads are discarded after training, the question remains: what output head for dense labels produces the strongest results?

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
