# OpenReview forum: "Multi Task Learning of Different Class Label Representations for Stronger Models"
_ICLR.cc/2023/Conference — Submitted to ICLR 2023_

### Official Review · Reviewer_tGTN · 2022-10-23

**Confidence:** 4
**Correctness:** 3
**Technical Novelty And Significance:** 2
**Empirical Novelty And Significance:** 3
**Recommendation:** 3

**Clarity, Quality, Novelty And Reproducibility:**

The paper is easy to read and understand. The technical novelties of the paper seems to be limited. The main contribution is to train a neural network architecture with an auxiliary task that uses Binary Labels as supervision. The paper does not clearly explain how the Binary Labels improves the learning. The multi-task learning architecture is basic with one independent branch for each task. I think a lot of implementation details (optimizer, learning rate, data preprocessing, number of epochs, batch size, etc) are missing so some experiments might be difficult to reproduce in the future.


**Strength And Weaknesses:**

**Strengths**
- The proposed auxiliary task seems to be quite easy to add to existing neural network architectures.
- The proposed approach increases the accuracy on five datasets: Cifar-100, Caltech101, Sun397, FGVC- Aircraft, and Stanford Cars. The proposed approach also seems to improve accuracy when the training is done with a subset of the training examples.

**Weaknesses**
- The technical contribution of the paper seems limited. The main idea is to use Binary Labels as an auxiliary task during training. The multi-task learning architecture is basic: one independent branch for each task.
- The paper should clearly explain how the Binary Labels are generated in the experiment section. There are multiple approaches that are presented in Section 4, but it is not clear which one is used in the experiment section.
- I am not sure to understand what is the motivation/intuition behind the Binary Labels. The Binary Labels do not bring semantic/meaningful information about the categories so why do they improve performances. I think the quality of the paper will improve if there was a paragraph to discuss about the properties that are modelled by Binary Labels but that are not modelled by one-hot labels. In other words, if Binary Labels is the solution, what is the problem?
- In the experiment section, it is important to report the standard deviation when the runs are averaged. It is important to understand the sensibility of the model to the random seed used to generate the Binary Labels.
- I wonder if data efficiency is a good metric for this paper because it is about label representation. I think that label efficiency may be a better metric. For a given label budget (i.e X labeled images and Y unlabelled images), what is the best performance? Data efficiency is maybe not a good metric because it is quite easy to collect a lot of images without labels, and then use a self-supervised learning algorithm to train a model.
- The proposed approach seems to be limited to multiclass image classification and does not seem to generalize to multilabel image classification.
- The proposed approach is not evaluated on large benchmarks like ImageNet. It will increase the quality of the paper to show the proposed approach generalizes to a large benchmark.
- There is no analysis of the weighting between the two tasks. It can be useful to add an analysis to show the robustness of the model to the weighting strategy.


**Summary Of The Paper:**

This paper introduces an approach to train a neural network for a multiclass image classification task. The main contribution is to add an auxiliary task, where Binary Labels are used as supervision. The Binary Labels can be carefully constructed or generated randomly based on the number of classes. The proposed approach is evaluated on five datasets: Cifar-100, Caltech101, Sun397, FGVC- Aircraft, and Stanford Cars. The proposed approach also seems to improve accuracy when the training is done with a subset of the training examples.


**Summary Of The Review:**

The technical contributions of the paper seems to be too limited. The main contribution is to train a neural network architecture with an auxiliary task that uses Binary Labels as supervision but it is difficult to understand why it improves performances. It seems there is no technical contribution on the multi-task learning part.

---

### Official Review · Reviewer_pCpT · 2022-10-24

**Confidence:** 4
**Correctness:** 4
**Technical Novelty And Significance:** 2
**Empirical Novelty And Significance:** 3
**Recommendation:** 3

**Clarity, Quality, Novelty And Reproducibility:**

- This paper is clearly written and easy to understand. The idea of using random binary partition of class labels as auxiliary tasks, while not very significant, is novel and effective.

Minor typos:
Lets -> Let's (section 3, 2nd paragraph)


**Strength And Weaknesses:**

Strengths:
- The method is simple and it works. The use of random binary partition of classes create useful related tasks for the neural network to train on. This might be important for more difficult classification tasks (e.g. cars and airplanes in the paper), where the sampled binary classification tasks can be easier than the original multi-class problem.

Weaknesses:
- The use of random binary partitions can be a strength but can also be a weakness, since unrelated classes can be grouped together. If class hierarchies are available (such as the ImageNet class hierarchy), they might improve upon the random binary partitions used in this work.

- While the main idea of the paper is simple and effective, the contribution might not be significant enough due to the small empirical improvements, and the lack of further analysis and insights into the problem of label encoding.

- Some classic works on multi-class label coding are missing, e.g.:
Dietterich, Thomas G., and Ghulum Bakiri. "Solving multiclass learning problems via error-correcting output codes." Journal of artificial intelligence research 2 (1994): 263-286.
Crammer, Koby, and Yoram Singer. "On the learnability and design of output codes for multiclass problems." Machine learning 47.2 (2002): 201-233.


**Summary Of The Paper:**

The authors propose improving the classification performance of neural networks by adopting a multi-task approach to label prediction.
In addition to the usual one-hot encodings, they consider random binary partitions of the classes and use them as classification tasks in a multi-task setting. Experiments show small improvements over baseline methods using standard one-hot encoding, especially for classification in special domains such as distinguishing different types of cars and airplanes.


**Summary Of The Review:**

The authors produce a piece of interesting work on using random binary partition of class labels as auxiliary tasks to improve the prediction accuracy of multi-class classification problems. While the result alone might not be significant enough for ICLR, the authors can consider expanding the results by exploring some of the questions they listed in the conclusions section.

---

### Official Review · Reviewer_gbMZ · 2022-10-24

**Confidence:** 4
**Correctness:** 1
**Technical Novelty And Significance:** 4
**Empirical Novelty And Significance:** 1
**Recommendation:** 1

**Clarity, Quality, Novelty And Reproducibility:**

Although this paper has novel aspects and reproducibility. Clarity and quality should be improved more.

**Strength And Weaknesses:**

The approach of improving performance by transforming labels is novel. This approach may be a direction that can improve performance in a way that is cheaper than data and network architecture improvements.

It is not easy to agree with the authors' claim that research on labels has been ignored. For example, knowledge distillation is a representative research direction that attempts to learn the more complex distribution of labels. Also, ordinal regression is labeled in a differentiated way to a one-hot vector. The authors should have discussed many studies related to labels.

The notations are generally poor, with unnecessary content and many calculation errors. The proposed algorithm is a trial without insight or analysis. For example, on page 3, it is difficult to understand what f_0 - f_{T-1}, h_0 - h_{T-1} mean. The three equations on pages 3 and 4 have little relation with the algorithm's core, and one of these equations is sufficient.
In the case of Chapter 4 on page 6, according to the authors' formula, when N=4, the number of groups becomes 10 instead of 7. This does not match the example in Figure 2.

Experimental results are insufficient to show the effectiveness of the algorithm. For most datasets, the baseline algorithm was set to those with too low performance. It is necessary to compare the SOTA method and the proposed algorithm.

**Summary Of The Paper:**

This paper proposes a method to improve image classification performance by modifying labels into various shapes and combining them. In each variant, the classes are separated 2-fold and labeled 0 and 1, respectively. Randomly selected combinations are used for classification training in combination with the existing softmax-based loss. In some limited experiments, this algorithm shows improved performance.

**Summary Of The Review:**

I think that the weaknesses must be sufficiently improved for this paper to be accepted.

---

### Official Review · Reviewer_2fNv · 2022-10-25

**Confidence:** 4
**Correctness:** 3
**Technical Novelty And Significance:** 3
**Empirical Novelty And Significance:** 3
**Recommendation:** 5

**Clarity, Quality, Novelty And Reproducibility:**

Clarity
=====

The paper discusses ‘dense labels’ in the introduction without explaining what they are, which makes it hard to follow that discussion.

The authors say they use Metabalance for multi-task learning but don’t explain what this means and don’t discuss it in the methods section.

The authors say “In this case, the two tasks are not just related, they are identical …”. This sounds odd since, in principle, it’s not clear why using the same task will be useful. Intuitively, if the tasks used for multi-task learning are redundant, training on them simultaneously won’t give more information than training on a single one.

The authors state that they randomly generate the binary labels for each class. Is this generation happening just once? I imagined that it would happen several times, each time resulting in a different set of classes being grouped together (in each embedding dimension), and I imagined that a different output head would be used for each. But in the experiments, it seems that there are a total of 2 output heads: one for the main task (softmax labels) and one for the (single) auxiliary task. It would be very useful to clarify the exact procedure that the authors propose. Perhaps including an Algorithm box too would be useful.

Novelty
======
The particular auxiliary training task proposed is novel, to the best of my knowledge. However, the authors should be more thorough in discussing related work (see comment below).

Reproducibility
============
I don’t think I’d be able to reproduce the results based on reading the paper. I’m not sure what Metabalance is and what is the modification that the authors propose to it. Some other details also weren’t clear to me (see Clarity section). I encourage the authors to make code available, and also to use an Appendix where all implementation details are described.

Quality
======
In Section 3, all loss equations sum over the training examples, whereas usually this is an average over the examples (ie. the term dividing by the number of examples is missing)

Figure 1 is low resolution, looks like it’s a screenshot. Please update to higher res.

The empirical evaluation section is weak. It would be useful to explore few-shot learning tasks at test time, robustness to distribution shifts, etc, as additional experiments. It would also be useful to compare against baselines, different variants of the proposed model (different ways of doing multi-task learning, etc) and compare against dense labels, as well as multi-task variants that include dense labels as an additional auxiliary task.

Since the task weighting seems to be important, it would be nice to do a study that shows how different values affect the results. Also, how was this hyperparameter determined for the main results in the paper?

The authors allude to results on using binary labels as the only or main task, but these results aren’t shown. This is puzzling as it seems there is plenty of space left in the paper.

Is the auxiliary task of binary labels complementary to that of dense labels? Can the two be combined?

If the random binary label generation happens exactly once (see comment in Clarity section), have the authors investigated the effect of that on performance? Maybe some random labels are better or worse than others.


Related work
==========
The authors mention this idea is similar to label smoothing, but they don’t discuss the differences aside from saying it is “quite different”. It would be useful to expand on that.

The proposed approach is closely related to unsupervised meta-learning, where tasks for meta-learning are created without using labels. One way of doing this is clustering, where hopefully different clusters correspond to different semantic classes, which can be used to form tasks (this clustering can be interpreted as defining a different labeling function) [1]. Another idea is randomly grouping examples into ‘classes’ so that different groups of examples get assigned the same label in each task [2]. This incentivizes the model to discover relationships between different subsets of the data in each task, which has a similar motivation to the binary labels introduced here.

Finally, a related idea is provided in [3] where meta-learning happens on tasks produced by different labeling functions. To create each task, a conjunction of attributes (like ‘smiling and wearing glasses’) is considered as a new label. This can also be thought of as a label transformation to create several tasks for multi-task learning.

References
=========
[1] Unsupervised Learning via Meta-learning. Hsu et al. ICLR 2019.

[2] Assume, Augment and Learn: Unsupervised Few-Shot Meta-Learning via Random Labels and Data Augmentation. Antoniou et al. 2020.

[3] Probing Few-Shot Generalization with Attributes. Ren et al. 2021.


Minor comments
==============
In section 2.1, “closes” → “closest”

Towards the bottom of page 8, “taks” → “task”


**Strength And Weaknesses:**

Strengths
========
[+] The paper proses an interesting idea and I agree that label representations are an understudied aspect which is orthogonal to where large gains usually come from (improving models and data)

[+] In some cases, the proposed method improves results significantly over vanilla training.

Weaknesses
===========
[-] In some cases, the performance improvement is very small compared to only training with softmax labels, and no intuition or analysis is provided to shed some light on when we expect this approach to help. For instance, is it for datasets with more finegrained labels? What are other factors that affect this?

[-] The experimental section is weak. It would have been useful to compare against additional baselines, perform more analyses, explore different datasets and scenarios (concrete details in the below section)

[-] The paper has some clarity issues.


**Summary Of The Paper:**

The paper proposes ‘binary labels’, where each dimension of the embedding space is assigned a 0 or 1 for each class, which is decided randomly at label construction time, before the start of training). Intuitively, at each embedding dimension, this leads to a grouping of classes (those that have a 0 in that dimension, and those that have a 1 there). So training to predict these labels can be thought of incentivizing the model to learn commonalities between different groups of classes. They propose to predict these binary labels as an auxiliary task, jointly with the main task of predicting the usual one-hot labels. They show that in some cases, both for in-distribution generalization and transfer learning, using this auxiliary task improves performance compared to the baseline of using only the main task. They also show that it can improve data efficiency (achieving higher accuracy with fewer updates) compared to that same baseline.

**Summary Of The Review:**

Overall, the authors propose a really interesting auxiliary task to co-train with the main task of predicting one-hot labels. The results are encouraging in some of the scenarios explored. However, I felt that this exploration is still too premature to meet the bar for publication. Additional baselines, analyses and experimental scenarios should be considered to gain more insights into the performance of the proposed approach. Clarity issues should also be addressed, and related work should be discussed more thoroughly.

---

### Decision · Program_Chairs · 2023-01-20

**Decision:**

Reject

**Justification For Why Not Higher Score:**

As listed in 'Weaknesses' above, reviewers raised a number of concerns. Among them, the most serious issue was that the novelty of the method was unclear. The authors did not submit the response, and the reviewers keep the ratings which are unanimous for rejection. The AC supports their opinions.

**Justification For Why Not Lower Score:**

N/A

**Metareview: Summary, Strengths And Weaknesses:**

Summary:
This paper proposed to modify labels into various shapes and utilize them in the form of multi-task learning to improve image classification performance. The core idea is to add auxiliary tasks in the network where the binary labels are used as supervision. The proposed approach is evaluated on five image classification datasets and shown to somewhat improve the performance from the standard cross entropy baseline.

Strengths:
1. The proposed method is simple and easy to implement but works.
2. The paper is generally easy to follow, although some clarity issues remain.

Weaknesses:
1. The novelty of the method is limited as label encoding and multi-task learning are both well-studied approaches.
2. The motivation/intuition and theoretical support behind the proposed Binary Labels are not clear.
3. Survey and discussion regarding related methods on label coding are not enough. Also, they are not compared in the experiment.
4. The experiments are rather weak. As the study in this paper is mostly empirical, more extensive comparison with other strong methods and baselines are needed.